# Mitochondrial Fusion Via OPA1 and MFN1 Supports Liver Tumor Cell Metabolism and Growth

**DOI:** 10.3390/cells9010121

**Published:** 2020-01-04

**Authors:** Meng Li, Ling Wang, Yijin Wang, Shaoshi Zhang, Guoying Zhou, Ruby Lieshout, Buyun Ma, Jiaye Liu, Changbo Qu, Monique M. A. Verstegen, Dave Sprengers, Jaap Kwekkeboom, Luc J. W. van der Laan, Wanlu Cao, Maikel P. Peppelenbosch, Qiuwei Pan

**Affiliations:** 1Department of Gastroenterology and Hepatology, Erasmus MC-University Medical Center, 3015 CE Rotterdam, The Netherlands; l.li.1@erasmusmc.nl (M.L.); l.wang.1@erasmusmc.nl (L.W.); s.zhang@erasmusmc.nl (S.Z.); g.zhou@erasmusmc.nl (G.Z.); b.ma@erasmusmc.nl (B.M.); j.liu.2@erasmusmc.nl (J.L.); c.qu@erasmusmc.nl (C.Q.); d.sprengers@erasmusmc.nl (D.S.); j.kwekkeboom@erasmusmc.nl (J.K.); w.cao@erasmusmc.nl (W.C.); m.peppelenbosch@erasmusmc.nl (M.P.P.); 2Department of Pathology and Hepatology, Beijing 302 Hospital, Beijing 100039, China; yijinwang927015@163.com; 3Department of Surgery, Erasmus MC-University Medical Center, 3015 CE Rotterdam, The Netherlands; r.lieshout@erasmusmc.nl (R.L.); m.verstegen@erasmusmc.nl (M.M.A.V.); l.vanderlaan@erasmusmc.nl (L.J.W.v.d.L.)

**Keywords:** mitochondrial dynamics, liver cancer, OPA1, MFN1

## Abstract

Metabolic reprogramming universally occurs in cancer. Mitochondria act as the hubs of bioenergetics and metabolism. The morphodynamics of mitochondria, comprised of fusion and fission processes, are closely associated with mitochondrial functions and are often dysregulated in cancer. In this study, we aim to investigate the mitochondrial morphodynamics and its functional consequences in human liver cancer. We observed excessive activation of mitochondrial fusion in tumor tissues from hepatocellular carcinoma (HCC) patients and in vitro cultured tumor organoids from cholangiocarcinoma (CCA). The knockdown of the fusion regulator genes, OPA1 (Optic atrophy 1) or MFN1 (Mitofusin 1), inhibited the fusion process in HCC cell lines and CCA tumor organoids. This resulted in inhibition of cell growth in vitro and tumor formation in vivo, after tumor cell engraftment in mice. This inhibitory effect is associated with the induction of cell apoptosis, but not related to cell cycle arrest. Genome-wide transcriptomic profiling revealed that the inhibition of fusion predominately affected cellular metabolic pathways. This was further confirmed by the blocking of mitochondrial fusion which attenuated oxygen consumption and cellular ATP production of tumor cells. In conclusion, increased mitochondrial fusion in liver cancer alters metabolism and fuels tumor cell growth.

## 1. Introduction

Primary liver cancer, including hepatocellular carcinoma (HCC) and intrahepatic cholangiocarcinoma (CCA), is one of the leading causes of cancer related death. This is mainly attributed to the global prevalence of hepatitis virus infections and the rising prevalence of non-alcoholic fatty liver disease [1]. Liver cancer is well-known for its resistance to classical chemotherapy, and therefore very limited therapeutic options are available [2]. Over the past decade, the research of cancer metabolism has largely extended our understanding of cancer biology and provided opportunities for therapeutic development [3].

Mitochondria, as double-membrane-bound organelles, are the cellular powerhouses in almost all eukaryotic organisms [4]. Metabolic reprogramming, a hallmark of cancer, is critical for the tumor initiation and development [5]. The Warburg hypothesis proposed that metabolism was altered due to mitochondrial defects, leading to an increase in glycolysis [3]. In addition to mitochondrial energetic metabolism transitions, the morphology of mitochondria is also dynamic in cancer [6,7]. Mitochondrial morphology is under control of the mitochondrial dynamics network, which continuously exists in cells and is comprised of fusion and fission processes [8]. Mitochondrial fusion, facilitated by Optic atrophy 1 (OPA1) and Mitofusin 1/2 (MFN1/2), is reported to be able to attenuate damage to mitochondrial DNA (mtDNA), proteins and lipids by mixing and dilution of mitochondrial matrix and membranes [9]. Therefore, larger and more efficient mitochondria may prevent excessive damage accumulation and better serve the metabolic needs of proliferating cells [10].

As the heart of the mitochondrial fusion process, OPA1 and MFN1/2 are respectively responsible for inner membrane and outer membrane fusion [11]. OPA1 plays a key role in mitochondrial cristae structure maintenance while cristae, invaginated by the inner membrane, are required for cellular adaptation to metabolic demand [12]. OPA1 deficiency in mice is lethal [13] and tissue specific depletions of OPA1 have demonstrated that mitochondrial fusion is vital for cellular metabolism [14]. OPA1 downregulation has also been reported to sensitize HCC cells to sorafenib treatment [15]. However, the detailed role of mitochondrial fusion in liver cancer remains largely elusive. In this study, we have investigated mitochondrial morphodynamics and its functional relevance in liver cancer.

## 2. Materials and Methods

### 2.1. Cells and Organoids Culture

Human HCC cell lines, Huh7 and SNU449, and human embryonic kidney epithelial cell line HEK 293T cells were maintained in Dulbecco’s modified Eagle’s medium (Lonza, Basel, Basel-Stadt, Switzerland) supplemented with 10% fetal bovine serum (Sigma-Aldrich, St. Louis, MI, USA) and 1% streptomycin/penicillin (Gibco, Thermo Fisher Scientific, Waltham, MA, USA). The obtained mycoplasma-free human cell lines were commercially checked monthly by GATC Biotech (GATC Biotech, Konstanz, Baden-Württemberg, Germany) and their identity verified by the molecular pathology department of the Erasmus MC as described previously [16]. Healthy liver, adjacent liver, and CCA organoids were cultured as previously described [17,18,19]. Informed consent was obtained from all patients and the use of patient materials was approved by the medical ethical committee of Erasmus Medical Center, Rotterdam (MEC-2014-060 and MEC-2013-143).

### 2.2. Gene Knockdown by Lentiviral Vectors

For gene knockdown, pLKO.1-based vectors targeting OPA1 and MFN1 and scramble were kindly provided by the Biomics Center in Erasmus Medical Center. Lentiviral-shRNA vectors packaging and infection were performed as described before [20]. Subsequently, Huh7 and SNU449 cells were selected by 3 µg/mL puromycin (Sigma-Aldrich, St. Louis, MI, USA) to generate stable knockdown cells. After selection, knockdown cells were cultured without puromycin for at least one month before use in further experiments. The generation of knockdown organoids was performed as described before [21].

### 2.3. Immunofluorescence of Live Cells and Frozen Tissues

We used tetramethylrhodamine (TMRM) to quantify the mitochondrial morphology of live cells, in a manner that is dependent on the maintenance of the mitochondrial membrane potential [22]. TMRM and Hoechst 33342 were purchased from Thermo Fisher.

Frozen tissues were sectioned at 8 µm and fixed in 4% paraformaldehyde. The stainings of the Voltage-dependent anion-selective channel 1 (VDAC1) and DAPI were performed as previously described [23]. Nonspecific staining was blocked by phosphate-buffered saline (PBS) supplemented with 5% serum, 1% bovine serum albumin and 0.2% Triton X-100 for 1 h. Samples were incubated with primary antibody VDAC1 (ab15895, Abcam, Cambridge, United Kingdom) diluted at 1:200 at 4 °C overnight and subsequently with goat anti-rabbit secondary antibody conjugated with Alexa Fluor 488 (BD biosciences, Franklin Lakes, NJ, USA) diluted at 1:1000 for 2 h at room temperature.

All images were acquired with a Zeiss LSM510META confocal microscope and quantified with ImageJ software (version 1.51, National Institute of Mental Health, Bethesda, Maryland, USA).

### 2.4. Colony Formation Assay and Alamar Blue Assay

Cells were seeded in 6-well plates (400 cells/well for Huh7 and 800 cells/well for SNU449), and cultured for 10 days. Formed colonies were washed with PBS and fixed by 3.7% formaldehyde for 10 min. Colonies were counterstained with Giemsa and their numbers were counted.

Organoid proliferation were evaluated for 3 days with an Alamar blue assay according to the manufacturer’s manual (Life Technologies, Thermo Fisher Scientific, Waltham, MA, USA).

### 2.5. Analysis of Cell Cycle and Cell Apoptosis

Cells were trypsinized and washed with PBS. After being fixed in cold 70% ethanol overnight at 4 °C, the cells were washed with PBS and incubated with 20 μg/mL RNase at 37 °C for 30 min. Subsequently, the samples with 50 μg/mL propidium iodide (PI) at 4 °C for 30 min, were analyzed by FACS for cell cycle.

Cell apoptosis analysis was performed by staining cells with Annexin V-FITC and PI as described before [24].

### 2.6. Xenograft Mouse Model in Nude Mice

The xenograft tumor model was established in 4–6 weeks female nude mice breeding in SPF environment. Mice were injected subcutaneously with knockdown cells and corresponding control cells into the lower left or right flank of the same mice (5 × 10^6^/200 μL cells per mouse; n = 6 mice per group), 1:1 mixed with matrigel. After the cell engrafting, tumor formation was monitored and measured on the first, seventeenth, nineteenth, twenty fourth and twenty seventh day. Then mice were sacrificed, and tumors were harvested and weighted. All animal experiments were approved by the Committee on the Ethics of Animal Experiments of the Erasmus Medical Center.

### 2.7. RNA Isolation and Sequencing

Cell lines and organoids were isolated according to manufacturer’s guidelines with total RNA isolation protocol (Invitrogen, Carlsbad, CA, USA). The quantity of total RNA was measured by a NanoDrop 2000 and the quality of total RNA was measured by Bioanalyzer RNA 6000 Picochip as a quality-control step. RNA sequencing was performed by Novogene with paired-end 150 bp (PE 150) sequencing strategy.

### 2.8. Real-Time PCR

Expression of mRNA in different samples were measured according to the manufacturer’s instructions [25]. GAPDH was considered as reference gene for normalization. The forward and reverse primers for OPA1 were as follows: TCAAGAAAAACTTGATGCTTTCA and GCAGAGCTGATTATGAGTACGATT. The forward and reverse primers for MFN1 were as follows: TGTTTTGGTCGCAAACTCTG and CTGTCTGCGTACGTCTTCCA.

### 2.9. Oxygen Consumption and ATP Production Measurement

The oxygraph-2k (O2k, OROBOROS instruments) was used for respiration measurements. Hepes/Tris buffer (adjusted to pH 7.4 with Tris) containing 4.2 mM KCl, 132 mM NaCl, 10 mM Hepes, 1.2 mM MgCl_2_ and 1 mM CaCl_2_, was used to incubate intact cells. The experiments were performed at 37 °C.

An ATP determination kit (Invitrogen, Carlsbad, CA, USA) was used for ATP measurement of cells. The final results were normalized by cell numbers or total protein concentration.

### 2.10. Measurement of Reactive Oxygen Species (ROS) Production

Cellular ROS Assay Kit (ab186027, Abcam) was used for ROS production measurements. The measurements were performed according to the manufacturer’s instructions. Cells were plated overnight in growth medium and harvested at 10^4^–4 × 10^4^ cells/100 µL per well. Further, 100 µL/well of ROS Red Working Solution was added into the cell plate and incubated at room temperature or in a 37 °C/5% CO_2_ incubator for one hour. Fluorescence activity was measured at Ex/Em = 520/605 nm (cut off 590 nm) with bottom read mode. The final results were normalized by cell numbers or total protein concentration.

### 2.11. Glucose Consumption Measurement

Glucose Assay Kit (ab65333, Abcam) was used for glucose level measurements. The measurements were performed according to the manufacturer’s instructions. Cells were plated overnight in growth medium and were harvested at 2 × 10^6^ cells/100 µL per well. Further, 100 µL/well of assay buffer was added into the cell plate. Cells were homogenized quickly by pipetting up and down a few times before being centrifuged for 2 min at 4 °C at top speed in a cold microcentrifuge to remove any insoluble material. Supernatant was collected and transferred to a clean tube. These enzymes were removed from sample by using Deproteinizing Sample Preparation Kit - TCA (ab204708, Abcam). Subsequently, 50 µL of reaction mix was prepared for each reaction. Cell plate was incubated at 37 °C/5% CO_2_ incubator for half an hour. The value was measured on a microplate reader at OD 570 nm. The final results were normalized by cell numbers or total protein concentration.

### 2.12. Lactate Measurement

A Lactate Assay Kit (ab65331, Abcam) was used for lactate level measurements. The measurements were performed according to the manufacturer’s instructions. The final results were normalized by cell numbers or total protein concentration.

### 2.13. Statistical Analysis

All data are presented as mean ± SEM. Statistical comparisons were performed with paired t test for paired samples or Mann-Whitney test for non-paired independent samples. For all experiments, a *p*-value less than 0.05 was considered as significant.

## 3. Results

### 3.1. Activation of Mitochondrial Fusion in Liver Cancer

Initially, we employed paired frozen tissues from thirteen HCC patients and one CCA patient by staining mitochondria with VDAC1, one of the most abundant proteins in the outer mitochondrial membrane. Details of the patient information were shown in Table 1. Of each patient, we included tumor tissue and adjacent tissue (Figure 1A). Immunofluorescence analysis showed that cells in tumor tissue and cells in adjacent tissue contained different mitochondrial morphology. In 10 out of 14 paired tissues mitochondrial volume per cell was larger in tumor tissue compared to adjacent tissue (Figure 1B). Interestingly, in all paired tissues tumor cells possessed stronger VDAC1 fluorescence intensity and integrated density (Figure 1C,D, respectively). We further examined mitochondrial morphology in paired HCC and adjacent tissue derived from an HBV positive patient by electron microscopy. Consistently, mitochondria size is larger in tumor tissue (Appendix A).

To investigate the clinical and functional relevance of mitochondrial morphodynamics in liver cancer, we examined key fusion proteins involved in mitochondrial dynamics. A previous study revealed that MFN1 mRNA levels are slightly elevated in liver while MFN2 is expressed at low levels in liver in contrast with heart and muscle [26]. Thus, our study mainly focused on OPA1 and MFN1. Analysis of the Oncomine online database, containing five cohorts of HCC patients, revealed that both OPA1 and MFN1 are more highly expressed in HCC tumor tissue compared to liver tissue in the majority of cohorts (Figure 1E,F).

We concluded that mitochondria in tumor and tumor surrounding tissues possessed different morphology and structural integrity.

### 3.2. Mitochondrial Fusion Sustains CCA Organoid Growth

We also observed a similar phenomenon in patient-derived CCA organoids (Figure 2A). One pair (up row) included normal organoids derived from the adjacent tissue and tumor organoids derived from the tumor tissue of the same patient. Another pair (lower row) included normal organoids derived from the donor liver and tumor organoids derived from another patient. Mitochondria in normal organoids are more sphere- or ovoid- like compared with the long filaments of mitochondria in tumor organoids. Further, 83.5% of organoids in tumor cells possessed elongated mitochondria, which is more than that in normal organoids (Figure 2B).

We consequently performed gene knockdown of OPA1 and MFN1 respectively in human CCA organoids and confirmed at both mRNA (Figure 2C) and protein levels (Appendix A). Consistently, this resulted in mitochondrial morphology transition (Figure 2D) and reduced mitochondrial length (Figure 2E). This functionally inhibited CCA organoid growth (Figure 2F).

### 3.3. Dysfunction of Mitochondrial Fusion Inhibits Liver Cancer Cell Growth In Vitro and In Vivo

To further investigate the underlying mechanism, we stably knocked down gene OPA1 or MFN1 by lentiviral transfection in two HCC cell lines Huh7 and SNU449. The successful knockdown was confirmed at mRNA (Figure 3A) and protein levels (Appendix A). The down-regulation of OPA1 or MFN1 induced mitochondrial fragmentation and a decrease of mitochondrial length, indicating the inhibition of mitochondrial fusion (Figure 3B,C). This functionally resulted in dramatic inhibition of ability of single cell colony formation (Figure 3D). The colony units of Huh7 shMFN1 dropped from average 71 per well to 20 and 9, and of Huh7 shOPA1 dropped from 31 to none. Similar results were observed in SNU449 cells.

Upon subcutaneous engraftment in immunodeficient nude mice, we demonstrated that knockdown of OPA1 or MFN1 dramatically inhibited tumor formation and growth of HCC cells in vivo (Figure 4A,B).

We found that knockdown of mitochondrial fusion regulators has no effect on cell cycle (Appendix A) but triggered the induction of cell apoptosis. This effect was much stronger in Huh7 shOPA1 cells compared to Huh7 shMFN1 cells. The percentage of apoptotic cells increased from 4.44% to 24.45% or 13.84% with the decrease of OPA1 gene expression, while the percentages of apoptotic cells increased from 3.17% to 4.79% or 4.64% with the decrease of MFN1 gene expression (Figure 5A,B). The apoptotic cells in SNU449 also showed the same trend with mitochondrial fusion dysfunction (Figure 5C,D). Interestingly, OPA1/MFN1 knockdown increased the ratio of Bax/Bcl-2 expression at mRNA levels in SNU449 and CCA organoids. This appears consistent with the observed increase of apoptosis in OPA1/MFN1knockdown cells (Appendix A). Taken together, these data suggest that dysfunction of mitochondrial fusion induces tumor cell apoptosis.

### 3.4. Mitochondrial Fusion Fuels Cellular Metabolism of Liver Cancer Cells

To obtain insight in the potential mechanisms that mediate acceleration of tumor cell growth by mitochondrial fusion in liver cancer, we performed genome-wide transcriptomic analysis by RNA sequencing in OPA1 knockdown Huh7. We identified 552 differentially expressed genes between the control cells and the shOPA1 cells (*p* < 0.05, log_2_Fc > 1), consisting of 332 genes downregulated and 220 genes upregulated (Figure 6A). KEGG pathway enrichment analysis revealed the alteration of several pathways, but most prominently of the metabolism pathway (Figure 6B), involving 33 differentially expressed genes (Figure 6C).

The dysfunction of mitochondrial fusion by OPA1/MFN1 knockdown was also found to be correlated to increased ROS levels in SNU449 and CCA organoids (Figure 6D,E). Furthermore, we performed the glucose consumption and lactate secretion measurement in SNU449 and CCA organoids (Appendix A). Glucose consumption was down-regulated when mitochondrial fusion was inhibited. The lactate level was also reduced when knocking down OPA1/MFN1, which indicated that glycolysis was not stimulated upon mitochondrial inhibition.

To provide further evidence for dysregulation of mitochondrial metabolism in cell lines, the mitochondrial metabolism of oxygen consumption and ATP production were measured in Huh7 (Figure 7A,B). The oxygen consumption rate (OCR) dropped from 37.68 pmol/s/10^6^ cells to 26.78 pmol/s/10^6^ cells and 21.63 pmol/s/10^6^ cells in Huh7 shOPA1 cells, while ATP content dropped to 79.27% and 76.22% compared with control group. Similar results were observed in Huh7 shMFN1 cells. Alterations of mitochondrial metabolism were also observed in CCA organoids, including a decrease in oxygen consumption (Figure 7C) and a reduction in ATP production reduction (Figure 7D). Therefore, enhanced mitochondrial fusion fuels cellular metabolism to functionally support liver cancer.

## 4. Discussion

Although the early Warburg hypothesis proposed glycolysis as the major metabolic process for energy production and anabolic growth in cancer cells due to mitochondrial defects, it recently became more clear that mitochondria also play key roles in oncogenesis [27]. In addition to the bioenergetic functions, mitochondria represent a cell signaling hub regulating cancer cell growth and fate [28]. The function of mitochondria is closely associated with their morphodynamics comprising of the fusion and fission processes. In this study, we found excessive activation of mitochondrial fusion in liver cancer, which provides a metabolic advantage to sustain tumor growth.

Hepatitis virus infections, namely HBV and HCV, are one of the leading causes of HCC. Mitochondrial fission has been frequently observed in HBV and HCV infections [29,30], which has been implicated in the attenuation of cell apoptosis. Whether this has an effect on malignant transformation towards the development of HCC remains unclear. In our HCC patients, we observed a clear enhancement of mitochondrial fusion in the tumor tissues compared to the adjacent liver tissues. Among all the tissues, only two pairs were HBV positive. Whether this discrepancy is attributed to the distinct etiologies of HCC remains to be further investigated.

In general, mitochondrial fission and its key regulator Drp1 has been widely studied in various types of cancers. Drp1 has been reported to be overexpressed in oncocytic thyroid tumors [31] but decreased in malignant colon and lung cancer tissues, whereas no change was observed in breast and prostate cancer [32]. Overall, fission is thought to be pro-tumorigenic, although it remains a matter of debate whether it has pro- or anti-apoptotic function [33]. Since, in our study, we found a clear elevation of mitochondrial fusion in HCC patient tissues and in vitro cultured tumor organoids from CCA patients, we thus focused on the functional relevance of this process. By genetic knockdown of the key fusion regulators OPA1 or MFN1, we inhibited mitochondrial fusion in HCC cell lines and CCA tumor organoids. This functionally inhibited cell growth in vitro and in vivo tumor formation in mice. This is in line with a previous study showing that the oncogene MYC increases membrane polarization and mitochondrial fusion in mammary epithelial cells to enhance cell proliferation [34].

We found that the inhibition of fusion did not affect cell cycling, but triggered apoptosis, pointing to an anti-apoptotic role of mitochondrial fusion in HCC. In this setting, it is likely the activation of the intrinsic apoptotic pathway, because no specific ligand was added. This intrinsic pathway is mainly triggered by non-receptor stimuli and characterized by the permeabilization of the outer mitochondrial membrane. This leads to the release of pro-apoptotic factors from the mitochondrial inter-membrane space into the cytosol [35]. Our results appear to be in line with the previous findings that mitochondrial fusion protein is responsible for maintenance of mitochondrial membrane structure, matrix homogeneity and mitochondrial genome integrity, which are vital for cell survival [36,37]. However, the exact mechanisms by which mitochondrial fusion modulates this apoptotic process remain to be investigated.

In MYC transformed cells, it has been demonstrated that fusion promotes mitochondrial metabolism to enhance cell proliferation [38,39]. Thus, we have performed genome-wide transcriptomic analysis and, consistently, the inhibition of fusion affected the metabolic pathway most predominantly. Functionally, we have demonstrated that the knockdown of fusion genes inhibited oxygen consumption and ATP production, the hallmarks of mitochondrial metabolism, in HCC cell lines and CCA tumor organoids. We also found the reduced glucose consumption and lactate secretion in fusion gene knockdown cells. In mouse embryonic fibroblasts, mitochondrial fragmentation impairs cell growth, showing widespread heterogeneity in mitochondrial membrane potential and suffered reduced respiratory capacity [40]. Although fission has been widely recognized as pro-tumorigenic, it has also been reported to decrease mitochondrial oxygen consumption rate and ATP production in malignant cells [41]. Cancer cells without mtDNA remain viable, although they are unable to form tumors, whereas reconstitution of oxidative phosphorylation again endows the tumorigenic capability [42]. Growing evidence demonstrates cross-talk between metabolic intermediates and ROS in cancer [43]. Mitochondrial dynamics and ROS processes influence each other in various cancers [44,45]. Our studies found that the dysfunction of mitochondrial fusion could increase ROS level in liver cancer cells. The morphodynamics and functions of mitochondria are rather multifaceted, highly depending on the cancer types and specific context. Thus, the distinct observations from different studies appear but may not be necessarily contradictive.

In summary, we found that the process of mitochondrial fusion is activated in liver cancer. This enhances mitochondrial metabolism to fuel tumor growth while resisting cell apoptosis. These findings bear important implications for understanding liver cancer biology and developing potential mitochondria-targeted therapy and more effective treatment modalities.

## Figures and Tables

**Figure 1 cells-09-00121-f001:**
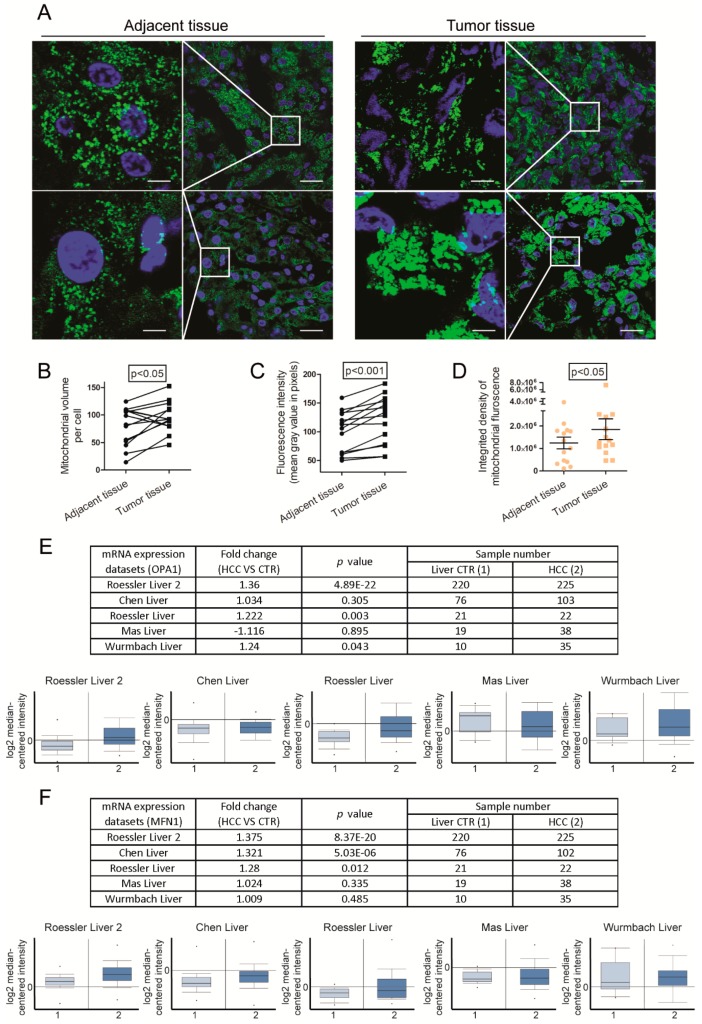
Activation of mitochondrial fusion in liver cancer. (**A**) Representative confocal images of mitochondria with VDAC1 staining in cryosection from paired tumor or adjacent liver tissues of two HCC patients. The white boxed regions were further magnified in the expanded images (left row). Scale bar = 5 µm (left row). Scale bar = 20 µm (right row). (**B**) Images of paired tissue were analyzed by ImageJ software. Mitochondrial volume per cell was determined by fluorescence area in pixels (Adjacent vs. Tumor: 80.29 ± 9.285 vs. 96.29 ± 7.24; n = 14 patients). (**C**) Fluorescent intensity represented the mean gray value of different images (Adjacent vs. Tumor: 100.3 ± 9.555 vs. 120.9 ± 11.18; n = 14 patients). (**D**) Integrated density was calculated by the product of area and mean gray value (Adjacent vs. Tumor: 1.245e+006 ± 260.203 vs. 1.848e + 006 ± 459.958; n = 14 patients). Histograms show means ± SEM with p value derived by two tailed paired Student’s *t* test. (**E**) The Oncomine microarray database (https://www.oncomine.org) was searched to analyze mRNA expression of OPA1 in HCC patients. In total, five cohorts of 423 HCC tumor tissues compared with 346 paired tumor-free tissues were identified. The expression level of OPA1 mRNA was demonstrated in five cohorts. (**F**) The Oncomine microarray database (https://www.oncomine.org) was searched to analyse mRNA expression of MFN1 in HCC patients. In total, five cohorts of 422 HCC tumor tissues compared with 346 paired tumor-free tissues were identified. The expression level of MFN1 mRNA was demonstrated in five cohorts. Histograms are mean ± SEM, with p values by Student’s *t* test.

**Figure 2 cells-09-00121-f002:**
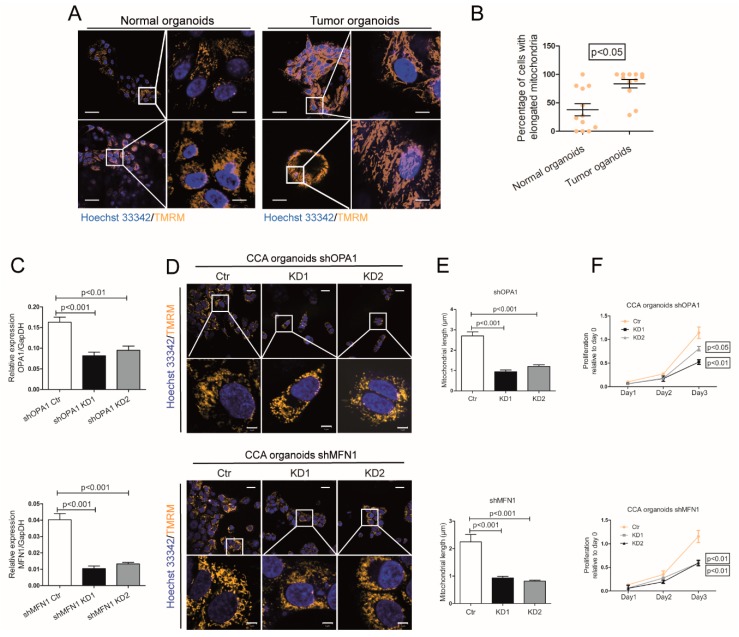
Silencing of OPA1 or MFN1 inhibits mitochondrial fusion and the growth of CCA organoids. (**A**) Representative confocal images of mitochondria with TMRM staining in in vitro cultured organoids from two tumor tissue, one matched adjacent liver tissue and one donor liver tissue. The white boxed regions were further magnified in the expanded images (right row). Scale bar = 50 µm (left row). Scale bar = 10 µm (right row). (**B**) Images of tumor and non-tumor “normal” liver organoids were quantified (Normal vs. Tumor: 37.93 ± 10.62 vs. 83.5 ± 7.34; n = 6 images/sample). (**C**) CCA organoids were transduced with mock lentivirus (Ctr) or shOPA1/shMFN1 lentivirus (KD1 and KD2) respectively. Gene knockdown efficiency of OPA1 was quantified by Real-time PCR (n = 6). (**D**) Representative live cell confocal images of mitochondria with TMRM staining in CCA organoids with OPA1/MFN1 knockdown. The white boxed regions were further magnified in the expanded images (lower row). Scale bar = 20 µm (upper row). Scale bar = 5 µm (lower row). (**E**) Quantified measurements of mitochondrial length in control and KD cells of CCA organoids (n = 3 images/sample). (**F**) The Alamar Blue fluorescence level of CCA organoids was measured at Day 0/1/2/3 for cell viability and data at day 0 was set as control of each group (n = 3). Histograms show means ± SEM with p value derived by the Mann Whitney test.

**Figure 3 cells-09-00121-f003:**
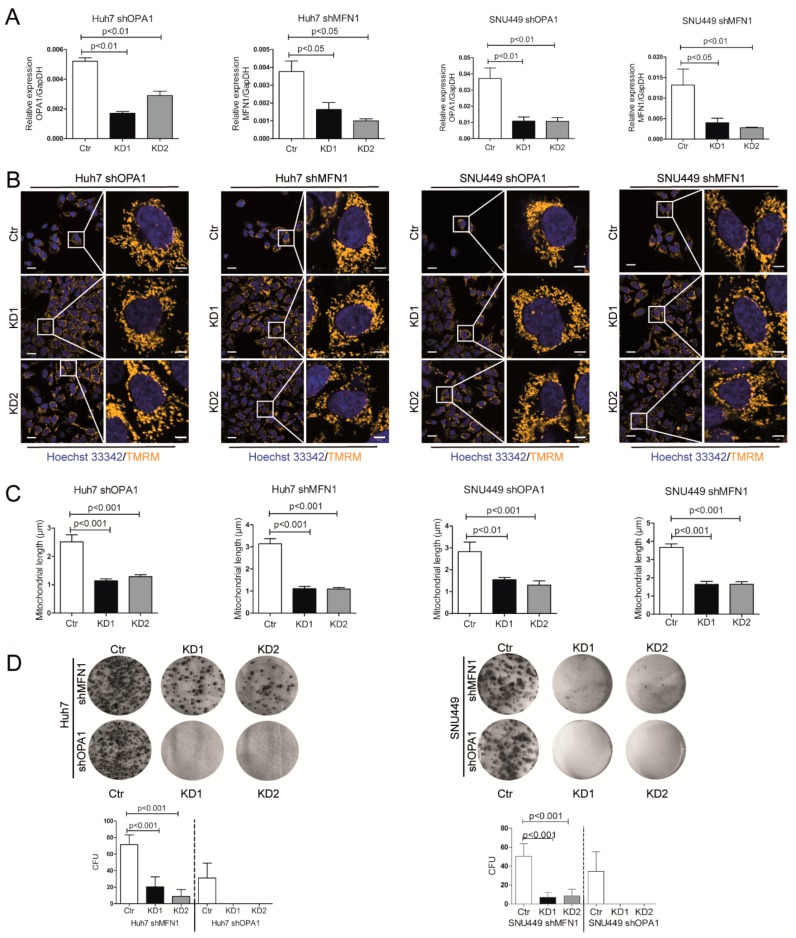
Knockdown of OPA1 or MFN1 inhibits mitochondrial fusion. Inhibition of mitochondrial fusion attenuates cancer cell growth in vitro. (**A**) Real-time PCR analysis of OPA1 expression in Huh7/SNU449 shOPA1 cells and MFN1 expression in Huh7/SNU449 shMFN1 cells. Huh7/SNU449 shOPA1 cells were transfected with mock lentivirus (Ctr) and shOPA1 lentivirus (KD1 and KD2) respectively. Huh7/SNU449 shMFN1 cells were transfected with mock lentivirus (Ctr) and shMFN1 lentivirus (KD1 and KD2) respectively. Gene knockdown efficiency was quantified (n = 6). (**B**) Representative confocal images of mitochondria with TMRM staining in live cells with OPA1 and MFN1 knockdown respectively. The white boxed regions were further magnified in the expanded images (right row). Scale bar = 20 µm (left row). Scale bar = 5 µm (right row). (**C**) Quantified measurements of mitochondrial length in control and KD cells (n = 3 images/sample). (**D**) Representative images of colony formation assay of Huh7/SNU449 shOPA1 cells and shMFN1 cells. Cells were fixed and stained by Giemsa (n = 9). The number of colony forming units (CFU) was quantified by ImageJ software. Histograms show means ± SEM with p value by Mann Whitney test.

**Figure 4 cells-09-00121-f004:**
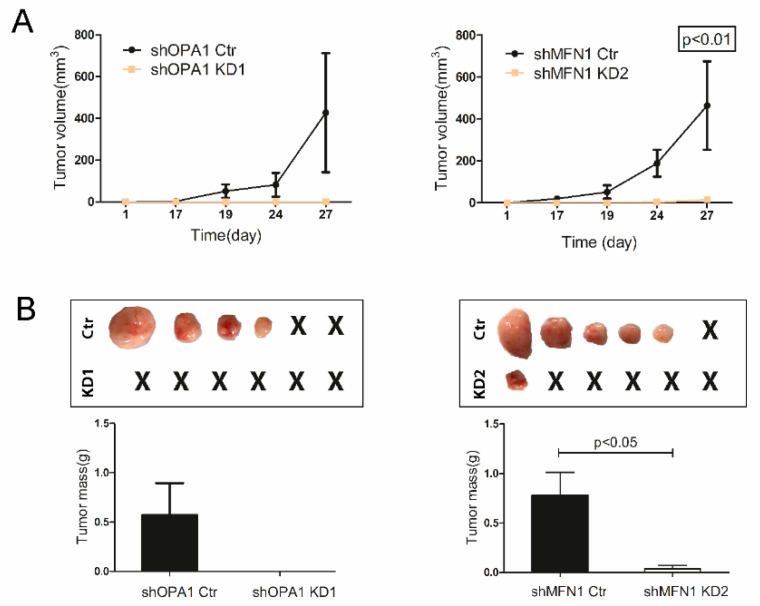
Mitochondrial fusion dysfunction inhibited tumor formation and growth in vivo. (**A**) Huh7 shOPA1 and control cells were injected subcutaneously into nude mice. Tumor size was measured at post-injection days 1/17/19/24/27 (n = 4 pairs). Huh7 shMFN1 and control cells were injected and tumor size was measured in the same way (n = 5 pairs). (**B**) Tumors were harvested from nude mice at day 27 and weighed. Histograms show means ± SEM with p value derived by the Mann Whitney test.

**Figure 5 cells-09-00121-f005:**
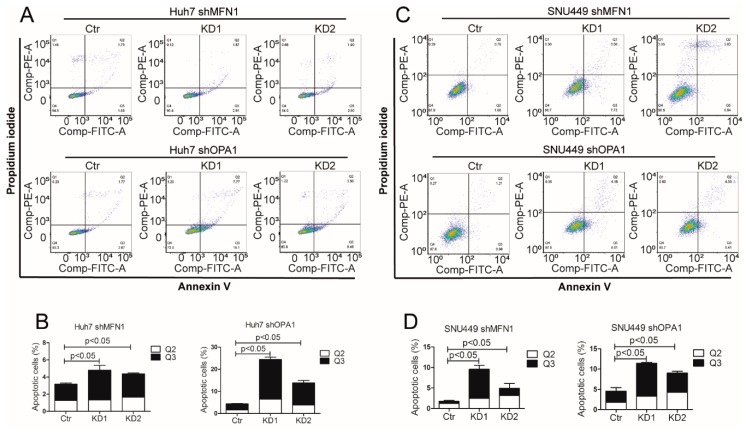
Mitochondrial fusion dysfunction inhibited tumor growth through cell apoptosis induction (**A**,**B**) Apoptotic cells were quantified by flow cytometry using Annexin V and propidium iodide co-staining in Huh7 shOPA1 and Huh7 shMFN1 cells (n = 3). (**C**,**D**) Apoptotic cells were quantified by flow cytometry using Annexin V and propidium iodide co-staining in SNU449 shOPA1 and SNU449 shMFN1 cells (n = 3). Histograms show means ± SEM with p value derived by the Mann Whitney test.

**Figure 6 cells-09-00121-f006:**
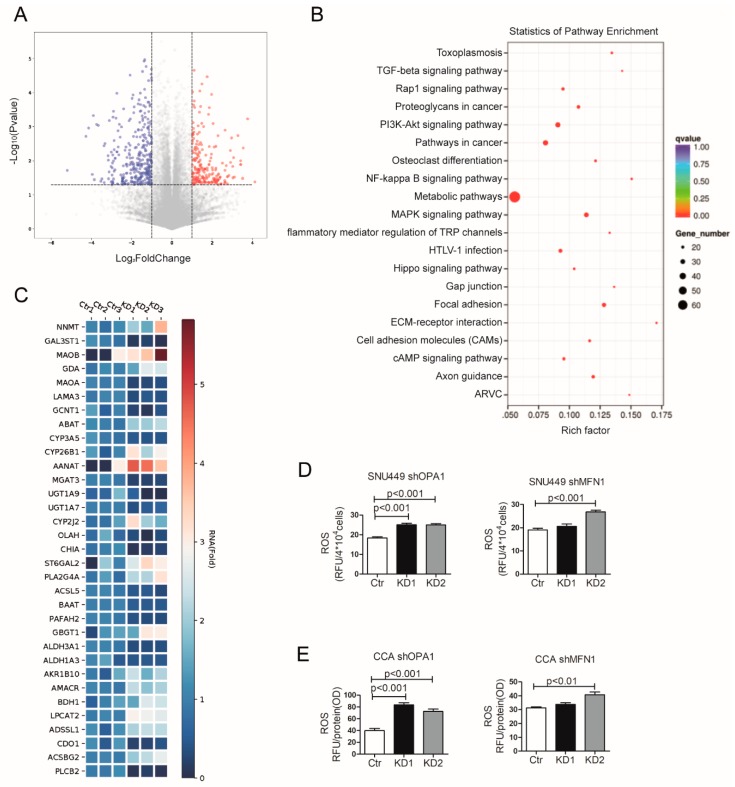
Inhibition of mitochondrial fusion affects cellular metabolism. (**A**) Volcano plot of statistical significance (*p* < 0.05) against fold change (ratio of KD/Ctr group), demonstrating the most significantly differentially expressed genes by genome-wide transcriptomic analysis between Ctr and shOPA1 Huh7 cells (n = 3). (**B**) KEGG pathway enrichment analysis within the complete set of differentially expressed genes (n = 3). (**C**) Heat map of color-coded expression levels of differentially expressed genes from metabolism pathway (two way ANOVA) (n = 3). (**D**) The ROS product level of SNU449 cells with OPA1/MFN1 downregulation showed increased ROS level compared with control group (n = 6). (**E**) The ROS product level of CCA organoids with OPA1/MFN1 downregulation showed increased ROS level compared with control group (n = 6). Histograms are mean ± SEM, with p values by Mann Whitney test.

**Figure 7 cells-09-00121-f007:**
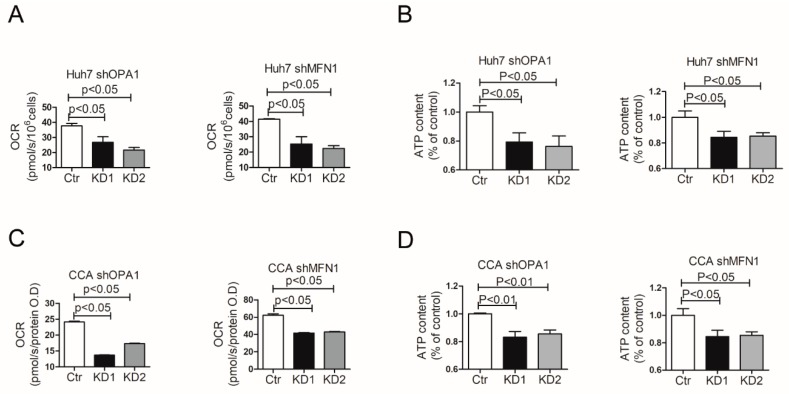
Inhibition of mitochondrial fusion affects Oxygen Consumption Rate and ATP production. (**A**) Real-time analysis of basal Oxygen Consumption Rate (OCR) in Huh7 shOPA1 and shMFN1 cells (n = 3). (**B**) ATP production of Huh7 cells with OPA1/MFN1 downregulation was reduced compared with control group (n = 6). (**C**) Real-time analysis of basal OCR in CCA organoids with shOPA1 and shMFN1 (n = 3). (**D**) ATP production of CCA tumor organoids with OPA1/MFN1 downregulation was reduced compared with control group (n = 6). Histograms show means ± SEM with p value derived by the Mann Whitney test.

**Table 1 cells-09-00121-t001:** Patient Characteristics.

Characteristics	HCC Patient (n = 13)	CCA Patient (n = 1)
Age at surgery (years)		
Mean ± SD	64.9 ± 13.4	60
Median (range)	70 (37–79)	-
sex (%)		
Male	8 (62)	1 (100)
Female	5 (38)	-
Etiology (%)		
Unknown liver disease	6 (46)	-
Alcohol abuse	1 (8)	1 (100)
Chronic HBV	3 (23)	-
Hemochromatosis	1 (8)	-
Hemochromatosis and alcohol abuse	2 (15)	-
Cirrhosis (%)		
Yes	2 (15)	-
No	11 (85)	1 (100)
Tumor differentiation (%)		
Well	1 (8)	-
Moderate	9 (69)	-
Poor	2 (15)	-
Unknown	1 (8)	1 (100)
Vascular invasion (%)		
Micro-invasion	6 (46)	-
Vaso-invasion	2 (15)	-
Severe vaso-invasion	2 (15)	1 (100)
No	3 (23)	-
AFP (pre-operative) (µg/L)		
Mean ± SD	4076.2 ± 12607.2	3.8
Median (range)	16 (3-45803)	-

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
