# Peer review of "Mitochondrial Fusion Via OPA1 and MFN1 Supports Liver Tumor Cell Metabolism and Growth"

_cells, 2020, doi:10.3390/cells9010121_

Round 1

Reviewer 1 Report

The authors have now addressed all my initial comments in the revised version of their manuscript. They have provided new sets of data that strengthen the study and help draw robust conclusions. I also strongly appreciate their attempt to silence OPA1/MFN1 in normal liver organoids, which is indeed technically very challenging. I have no further comment and I now recommend the manuscript for publication. 

Reviewer 2 Report

This revision is better can consider for publish.

This manuscript is a resubmission of an earlier submission. The following is a list of the peer review reports and author responses from that submission.

Round 1

Reviewer 1 Report

Dear,

I have now carefully read and examined the manuscript entitled “Mitochondrial fusion fuels liver tumor cell metabolism and growth” by Li et al. The authors provide some new insights on how mitochondrial fusion, in particular OPA1 and MFN1 protein components, may support liver cancer growth and progression. Although this study is of potential interest for the research field, it requires some revisions before being further considered for publication.

More precisely, the authors must show the effects of OPA1 and/or MFN1 KD in normal hepatic cells and/or normal organoids. Moreover, the authors should at least discuss whether mitochondrial fusion is known to correlate with tumor stage. Is there any correlation between OPA1/MFN1 expression and tumor invasion status?

It is unclear which energetic substrates were used to fuel mitochondrial respiration during OCR measurements. Glucose, glutamine, fatty acids? Is there any preference for either substrate when mitochondrial fusion is activated? Please comment and/or provide experimental data.

Is glycolysis stimulated upon mitochondrial fusion inhibition? Please add new experimental data on ECAR values in cells with shOPA1 or shMFN1 as well as glucose consumption and lactate secretion.

Minor comments:

The title should be edited as follows “mitochondrial fusion supports liver tumor cell metabolism and growth”.

Reviewer 2 Report

The authors that examined the Optic atrophy 1 (OPA1) and Mitofusin 1 (MFN1) function on mitochon

The authors that examined the Optic atrophy 1 (OPA1) and Mitofusin 1 (MFN1) function on mitochondria fusion. The mitochondria fusion is correlated to liver cancer growth, which also required for ATP synthesis and ROS metabolism. This topic is interesting, but have a little complex, did not just said some things for enough to cover. Some comments is as following:  

In the title is not clear for the reader that may be modulated as “Mitochondria fusion via OPA1 and MFN1 can regulate liver cancer cell growth and metabolism in vitro and in vivo.”

In the figure 1 should be checked the ROS production that super fission in mitochondria is correlated to a lot of ROS increasing. On the other hand, mitochondria dye MitoSox Red should be used in the staining.

In the figure 2, OPA1 and MFN1 expression level should be checked by western blot analysis during knockdown by shRNA.

In figure 5, in the apoptosis assay should check more factors on cell death such as Bax, p53 and caspase-3 et al.

In the Discussion, should discussed more on why mitochondria fusion can reduce the ROS production.

6. The energy provided from mitochondria, but minor generation ATP from autophagy process, especially from mitophagy, if it possible autophagy markers should be checked.  dria fusion. The mitochondria fusion is